# Identification and Characterization of *Colletotrichum* Species Causing Sorghum Anthracnose in Kenya and Screening of Sorghum Germplasm for Resistance to Anthracnose

**DOI:** 10.3390/jof9010100

**Published:** 2023-01-11

**Authors:** Irene Njeri Koima, Dora Chao Kilalo, Charles O. Orek, John Maina Wagacha, Evans N. Nyaboga

**Affiliations:** 1Department of Plant Science and Crop Protection, University Nairobi, P.O. Box 29053, Nairobi 00625, Kenya; 2Department of Agricultural Sciences, South Eastern Kenya University, P.O. Box 170, Kitui 90200, Kenya; 3Department of Biology, University Nairobi, P.O. Box 30197, Nairobi 00100, Kenya; 4Department of Biochemistry, University Nairobi, P.O. Box 30197, Nairobi 00100, Kenya

**Keywords:** anthracnose, *Colletotrichum sublineola*, internal transcribed spacer, resistance, sorghum

## Abstract

Anthracnose caused by *Colletotrichum* species is one of the most destructive fungal diseases of sorghum with annual yield losses of up to 100%. Although the resistance to anthracnose has been identified elsewhere, the usefulness of the resistance loci differs depending on the pathogen species and pathotypes. Accurate species identification of the disease-causing fungal pathogens is essential for developing and implementing suitable management strategies. The use of host resistance is the most effective strategy of anthracnose management and therefore identification of sources for resistance against unique pathogen pathotypes is fundamental. The aims of this study were to identify and characterize *Colletotrichum* species associated with sorghum anthracnose and screen sorghum germplasm for resistance to anthracnose. Symptomatic sorghum leaf samples were collected from smallholder farmers in lower eastern Kenya and used for the isolation, identification and characterization of *Colletotrichum* species using morpho-cultural and phylogenetic analyses with the sequences of the rDNA internal transcribed spacer (ITS) region. Pathogenicity tests of the seven fungal isolates showed that there were no significant differences in the pathogenicity on host plants. The fungal isolates were variable in cultural and morphological characters such as colony type and color, colony diameter, mycelia growth and hyaline. The phenotypic characters observed were useful in the identification of the genus Colletotrichum and not the species. Based on the sequence and phylogenetic analysis of ITS, *Colletotrichum sublineola* was revealed to be associated with anthracnose on sorghum. Germplasm screening for resistance to anthracnose showed differential reactions of sorghum genotypes to anthracnose under greenhouse and field conditions. The results revealed four resistant genotypes and ten susceptible genotypes against *Colletotrichum sublineola*. Significant (*p* ≤ 0.05) differences were observed in grain weight, grain yield, weight of 100 seeds and harvest index among the tested sorghum genotypes. The present study indicated that the Kenyan accessions could be an important source of resistance to anthracnose. The findings from this study provide a platform towards devising efficient disease control strategies and resistance breeding.

## 1. Introduction

Anthracnose caused by *Colletotrichum sublineola* is the most damaging foliar fungal disease of sorghum especially in the tropical and subtropical regions where the conditions are favourable for disease development [1]. The disease is prevalent in all sorghum growing areas of sub-Saharan Africa where it causes huge economic losses in terms of quality and quantity of grain yields, depending on the susceptibility of the sorghum cultivars, epidemics of the region and environmental conditions [2]. Yield losses of up to 67% have been reported in susceptible cultivars and areas characterized by alternating dry and humid weather conditions combined with moderate to high temperatures [3]. It is now widely believed that sorghum anthracnose, caused by the *Colletotrichum* species, can cause damage throughout the cultivated period of sorghum. When sorghum anthracnose occurs, all tissues of sorghum can be infected. Therefore, without efficient disease management practices, the disease can devastate the whole crop [4].

Anthracnose disease in sorghum occurs in four phases including root rot at seedling stage, leaf and sheath phase, stalk rot and grain mold [5]. The phases can occur within a single growing season. Root rot is caused by planting infected seeds or contaminated soil, and the foliar phase appears 30–40 days after plant establishment and develops quickly close to the end of vegetative growth and at the beginning of the booting stage, while grain mold occurs during grain filling to physiological maturity [6]. The symptoms associated with anthracnose include circular and/or elliptical spots which elongate and coalesce covering the entire leaf with the presence of few or many fungal fruiting bodies known as acervulus within leaf lesions, stalk rot, peduncle breakage and grain deterioration [7]. Foliar infections occur at any stage of plant development and are reported to cause yield losses of up to 80% through the reduction of kernel size and number and grain filling [6,8]. Infected grains appear dark brown or black with streaks encircling the grains reducing their quality. Additionally, when an infected seed is sown it may result in the reduction of germination and the introduction of new *C. sublineola* pathotypes in new regions [9]. Naturally *C. sublineola* fungus occurs in mitosporic form and survives as conidia, mycelium, and sclerotia in crop debris, wild hosts and infected seeds [5,10].

The accurate diagnosis of *Colletotrichum* species causing anthracnose is important due to the differences in pathogenicity-related characteristics among species [11]. Identification of the causal agent(s) facilitates the development of effective management and control strategies and resistant breeding programs for anthracnose in sorghum [12]. *Colletotrichum* species are highly variable as manifested by morphology of the colony, shape of conidia, presence and shape of setae and appressoria, pigmentation, pathogenicity and other traits [13,14]. The identification of *Colletotrichum* species has been traditionally carried out based on the host range, cultural and morphological characters [15]. However, due to the plasticity of these characteristics and their propensity for variation under different environmental conditions, such morphology-based identification methods are not reliable for taxonomic demarcation [12,16,17,18,19,20,21]. A combined morpho-cultural with sequence data can be used to circumvent the challenges of using traditional methods of species identification [15,19]. Sequence comparisons of different metabolic and structural genes including the ITS region have proven useful for studying the phylogenetic relationships of *Colletotrichum* species [15,18]. The identification of the pathogenic *Colletotrichum* species causing sorghum anthracnose through sequencing of ITS and phylogenetic analysis is yet to be undertaken in Kenya.

Many strategies have been employed to control anthracnose including cultural practices, fungicides and use of resistant germplasm. Utilization of resistant germplasm is the most effective, economical and eco-friendly strategy for sorghum anthracnose management [22,23]. Several African and other countries have made tremendous efforts to evaluate sorghum resistance to anthracnose [24,25,26,27,28,29]. For example, 25 sorghum accessions from Zimbabwe have been identified as resistance to anthracnose [24]. In addition, 12 sorghum accessions from Burkina Faso and South Africa were shown to have resistance to anthracnose [25]. However, there are over 40 variable pathotypes of *C. sublineolum* that have been reported from different geographical regions that break resistance of the identified sorghum genotypes over time [30]. This necessitates the identification of region-specific and additional sources of anthracnose resistance for the development of new resistant sorghum cultivars. Screening of local and improved genotypes is important to reveal new sources of *C. sublineolum* resistance that could be incorporated into breeding program [31].

The aims of this study were to (1) identify and characterize *Colletotrichum* species associated with sorghum anthracnose in Kenya based on morphology and phylogenetic analysis using ITS sequence data; (2) evaluate the response of local and improved sorghum germplasm to anthracnose under greenhouse and field conditions; and (3) identify possible resistance sources to *Colletotrichum* species that can be used in future breeding programs. This is an important consideration in resistance breeding programs and disease control management including selection of cultivars for cultivation, and the selection of appropriate isolates to screen for resistance in plant breeding programs.

## 2. Materials and Methods

### 2.1. Collection of Diseased Sorghum Leaf Samples, Isolation and Purification of Fungi

A total of twenty-six sorghum leaf samples were collected during the short growing season of February and March 2019, of which 10, 9 and 7 samples were obtained from Kitui, Machakos and Makueni Counties, respectively (Appendix A). The leaf samples were obtained during a survey for fungal foliar and panicle diseases infecting sorghum. The samples were collected from leaves with typical anthracnose symptoms, characterized by small to large spots with a white center surrounded by red to black boarders. The spots were elongated in some leaves and coalesced forming larger spots in the entire leaf. Some parts of the leaf sheath and stalk were also colored with red to brown lesions.

The fungal pathogens were isolated following the method described by Kutawa et al. [32] with slight modifications. Lesion margins on sorghum leaves were aseptically excised into 3 mm × 3 mm, surface sterilized with 1.3% sodium hypochlorite and rinsed thrice with sterile distilled water. Sterilized tissue segments were placed on sterile blotting paper, allowed to dry under laminar flow, transferred aseptically and cultured on potato dextrose agar (PDA) and kept at 25 ± 2 °C for 3–7 days in the dark. Newly emerging mycelia were transferred to fresh PDA. Pure cultures were preserved in 15% glycerol at −80 °C, for further use.

### 2.2. Pathogenicity Tests of Fungal Isolates

Pathogenicity tests of the seven fungal isolates were conducted on detached leaves of sorghum cultivar Kateng’u. Healthy leaves were obtained from sorghum plants growing in pots in a greenhouse. The leaves were washed with tap water and then submerged in 1.3% sodium hypochlorite for three minutes followed by rinsing three times with sterile water. The sterilized leaves were air-dried on sterile paper towels under a laminar hood. After drying, the leaves were cut and placed in a 90 mm-diameter Petri dishes lined with two layers of sterilized blotting papers. The blotting papers were moistened with sterile distilled water to maintain high humidity (>95%). The leaves were inoculated using the wound/drop inoculation method. The leaves in the Petri plates were wounded by pin-pricking the upper surface with sterilized needle (insect pin, 0.5 mm diameter) and inoculated with 1 × 10^5^ conidia/mL concentration of spore suspension prepared from each isolate and left at 25 ± 2 °C for 7 days [33]. Healthy leaves were mock-inoculated with sterile distilled water, which served as the control. After 10 days, the pathogenic fungal pathogens were re-isolated from the inoculated symptomatic leaves and identified by morphological characteristics and molecular sequencing using the methods described in the subsequent sections and the isolates were compared to the original cultures [33].

### 2.3. Morpho-Cultural Characterization

Cultures grown on PDA at 25 ± 2 °C were used for morphological analysis as described by Kimaru et al. [34] and Kutawa et al. [32]. Mycelial plugs (5 mm diameter) were cut from margins of a 3-day-old actively growing colony and placed on fresh PDA in 9 cm-diameter Petri dishes with four plates per fungal isolate. The isolates were kept and incubated at 26 ± 2 °C for 10 days.

Colony diameters were recorded at 24, 48, 72, 96, 120, 144 and 168 h and mean colony growth determined. The mycelia color on PDA and growth pattern was also recorded. Conidia production, size, color and shape were examined up to three weeks of culture on PDA incubated at 25 ± 2 °C following the slide culture techniques described by Kutawa et al. [32]. Approximately 10 mm^2^ of PDA were placed on the slide, edges of agar inoculated with spore from sporulating culture and covered with a slip. Inoculated slide culture was placed in a sterile Petri plate containing a wet bottling paper sealed with parafilm and incubated at room temperature for 5 days. The slide cultures were observed under Lx400 compound microscope (LaboMed, Beijing, China) and mean length and width of at least 30 spores measured using calibrated ocular slide and stage micrometer. Appressoria were produced from hyphae using slide culture technique described by Kimaru et al. [34]. Seven fungal isolates were morphologically identified as *Colletotrichum sublineola* and were used for subsequent molecular identification and characterization.

### 2.4. DNA Extraction, Polymerase Chain Reaction (PCR) and Sequencing

To extract genomic DNA, pure cultures of seven *Colletotrichum* isolates (initially identified on the basis of morpho-cultural characteristics) were incubated on PDA at 25 ± 2 °C for 7 days. After 7 days of growth, mycelia were harvested by scrapping the surface with autoclaved spatulas and DNA was extracted using cetyltrimethylammonium bromide (CTAB) method [35]. PCR amplification was carried out with two sets of primers, the *Colletotrichum* specific primer, [Col-1 F (5′-AAC CCT TTG TGA ACR TAC-3′) and Col-2 R (5′-TTA CTA CGC AAA GGA GGC T-3′)] and the internal transcribed spacer (ITS) primer [ITS1 (5′-TCC GTAGGT GAA CCT GCG G-3′) and ITS4 (5′-TCCTCC GCT TATTGA TAT GC-3′)] as described by White et al. [36]. PCR amplifications were done in 50 µL reaction volume, containing 25 µL master mix (Horse-power^TM^ Green-Taq DNA polymerase 2×), 22 μL of nuclease free water, 1 μL of forward and reverse primer and 1 μL of DNA template.

The thermal cycling conditions were: initial denaturation at 94 °C for 5 min followed by 35 cycles of denaturation, annealing and extension at 94 °C for 30 s; 52 °C for 40 s (for Col) or 54 °C for 40 s (for ITS); 72 °C for 30 s and a final extension at 72 °C for 5 min. The amplified products were checked by 1.5% agarose gel electrophoresis and compared to 1 kb DNA ladder (DNA Ladder Mix 1 kb, 1st Base Company, Singapore). The amplified products for ITS primer were sequenced by Macrogen Europe BV Laboratory (Amsterdam, The Netherlands) using an Applied Biosystems 3730XL DNA Analyzer platform.

### 2.5. Sequence Alignment and Phylogenetic Analysis

The sequences from forward and reverse primer sequencing were assembled, manually edited, trimmed and aligned with Molecular Evolutionary Genetic Analysis (MEGA) version 11. The consensus sequences were subjected to the Basic Local Alignment Search Tool (BLAST) search (https://blast.ncbi.nlm.nih.gov/Blast.cgi; accessed on 5 July 2022) against the nucleotide database in the UNITE fungal database and NCBIs GenBank nucleotide database to determine closest relative species of *Colletotrichum* to the seven isolates under study. The sequences of the isolates in this study were deposited in GenBank, and the accession numbers assigned.

Phylogenetic analysis for the sequences was done by Molecular Evolutionary Genetic Analysis (MEGA) Version 11. ITS sequences of the isolates and sequences from different species of *Colletotrichum* retrieved from GenBank were used to construct the phylogenetic tree. *Glomerella acutata* and *Colletotrichum acutata* were used as out-group species. A neighbour-joining (NJ) analysis was performed individually for each sequence using the Maximum Composite Likelihood method and 1000 bootstrap replications to determine the statistical support of the phylogeny. Unweighted Pair Group Method with Arithmetic (UPGMA) mean was used to generate a nearly identical topological tree in MEGA. Evolutionary distances inform of number of bases substitution per locus were computed by maximum composite likelihood method [37]. The resulting phylogenetic tree was viewed by the fig tree software.

### 2.6. Screening of Sorghum Germplasm for Resistance to Anthracnose

The experiments were carried out under greenhouse and field conditions. Fourteen (14) sorghum genotypes (5 local and 9 improved) were obtained from International Crops Research Institute for the Semi-Arid Tropics (ICRISAT), Kenya and evaluated for their reaction to *Colletotrichum sublineola*. The local genotypes were Kaguru, Rasta, Kateng’u, Mugeta, and Kauwi while improved were Serena, Seredo, KARI Mtama-1, Makueni Local-2, Kiboko Local-2, Marcia, Gadam, Dark red and IESV 24,029 SH. Kaguru and Kateng’u were used as susceptible controls for both field and greenhouse experiments. Certified seeds were used for all the 14 sorghum genotypes used in this study.

#### 2.6.1. Greenhouse Screening Trials

Eight (8) seeds of each sorghum genotype were planted in 30-cm diameter pots containing sterile soil in the greenhouse at 28 ± 2 °C. After germination, the plants were thinned to 5 plants per pot. The experiment was conducted in a randomized complete block design with three replicates of five plants per replication. Thirty-days-old plants were inoculated with *C. sublineola* (Isolate KT001-A), prepared following the method described by Ramathani et al. [38] using a hand-held atomizer, allowed to dry for 10 min, then covered with a polythene bag to maintain 95% relative humidity [39]. Three fully mature healthy sorghum leaves were sprayed for each of the five plants used per replication. The control plants were mock-inoculated by spraying with sterile distilled water and maintained in the same conditions. The polythene bags were removed after six days and the plant left under greenhouse conditions. The severity of anthracnose disease symptoms and reaction types [resistant (R), moderately resistant (MR), moderately susceptible (MS), susceptible (S) and highly susceptible (HS)] were estimated by mean percentage leaf area covered by the disease, and the ratings were carried out using the standard scale (1 to 9) as described by Thakur et al. [40] (Table 1). Scores were taken on a weekly basis starting from seven days to 12 weeks post-inoculation or when the highly susceptible genotype succumbed to the disease. The disease severity data was subjected to analysis of variance using the Statistical Analysis System software version 9.2 [41] and means separated by least significant difference. Four sets of growing periods were used to generate Area Under Disease Progressive Curve (AUDPC), which includes 1–21, 22–42, 43–63 and 64–84 weeks post-inoculation. The AUDPC was calculated from the severity data following the method proposed by Madden et al. [42] using the following equation,
$$AUDPC=\sum_{i=0}^{n}\left[\left(Xi+Xi+1\right)/2\right]\left[ti+1-t1\right]$$

In the formula, *Xi* and *Xi* + 1 is the percentage of leaf area infected with the disease at *i*th and *i*th + 1 observations respectively, *ti* + 1 and *ti* is the days (time) at *i*th +1 and *i*th observations respectively and *n* refer to the total number of observations.

Twelve weeks post-inoculation the chlorotic lesions were collected, *C. sublineola* re-isolated and compared with the original isolate used for inoculation.

**Table 1 jof-09-00100-t001:** Assessment scale of anthracnose disease against sorghum plants under greenhouse conditions.

Severity Ratings	Symptoms and Lesion Type on the Leaves	DRC
1	0 to <1% leaf area covered with hypersensitive reaction with mild yellow flecks	R
2	1–5% leaf area covered with hypersensitive lesions without acervuli	MR
3	6–10% leaf area covered with hypersensitive lesions without acervuli	MS
4	11–20% leaf area covered with hypersensitive and restricted necrotic lesions with acervuli	
5	21–30% leaf area covered with hypersensitive and restricted necrotic lesions with acervuli	S
6	31–40% leaf area covered with coalescing necrotic lesions with acervuli	
7	41–50% leaf area covered with coalescing necrotic lesions with acervuli	HS
8	51–75% leaf area covered with coalescing necrotic lesions with acervuli	
9	76–100% leaf area covered with coalescing necrotic lesions with acervuli	

The assessment scale is adopted from Thakur et al. [40] and Xu et al. [43]. DRC—Disease Reaction Classes: Resistant plants (R) were rated as 1, moderately resistant plants (MR) as 2, moderately susceptible (MS) as 3 and 4, susceptible plants (S) as 5 and 6 and highly susceptible plants (HS) as 7, 8 and 9.

#### 2.6.2. Field Screening Trials

Field screening trials were conducted in the short season (November to March) of 2020 and long season (April to August) of 2021 at the Kenya Agricultural and Livestock Research Organization (KALRO) Kiboko and Ithookwe field stations. These are high-pressure disease areas for sorghum anthracnose disease due to their hot and relatively humid conditions [44]. Kiboko field station is located at Makindu sub-County, lies at a longitude of 2°15′ S and a latitude of 37°75′ E, an altitude of 975 m and has an average temperature of between 14.3 °C and 35 °C, with an annual rainfall of 530 mm. Ithookwe field station is located at Kitui rural sub-County along 1°22′34′′ S and 37°58′43′′ and an altitude of 1147 m above sea level [45].

Seeds of the 14 sorghum genotypes were sown at the onset of rains at seed rate of 8 kg/ha. The genotypes were randomly sown in 0.60 m rows with plots of 4 × 2.5 m in a randomized complete block design (RCBD) with four replicates. Each plot had four rows and blocks were separated by 1.5 m path. Sorghum cultivar “Sila” was planted at the guard row and Wagita at every 9th row in the field to serve as spreader rows. The 14 genotypes were sown between the spreader rows 21 days later when foliar symptoms were fully visible in the spreader rows. Di-ammonium phosphate (DAP) was applied at planting at a rate of 40 kg/ha and top dressing was done with calcium ammonium nitrate (CAN) at rate of 30 kg/ha when the sorghum crop produced eighth leaf. Weeding was done thrice before the crop maturity at each season. An insecticide, Duduthrin (1.75 g/L) was applied three weeks after emergence to control shoot fly and stem borer at the rate of 200 mL/ha and bird guarding initiated as soon as seeds were formed in the panicles to prevent damage of grains. Data on anthracnose disease severity, grain weight, weight of 100 seeds and grain yield were collected.

The severity of anthracnose disease symptoms and reaction types (R, MR, MS, S and HS) were estimated by the percentage leaf area covered by the disease using a standard scale (1 to 5) as described by Thakur et al. [40] (Table 2). Ten plants on each of the two middle rows of plots were randomly selected and tagged to enable scoring of anthracnose disease symptoms. Data on anthracnose disease severity was collected from the tagged plants in each plot at 45 (knee high), 75 (booting stage), 95 (50% flowering) and 105 (physiological maturity) days after sowing (DAS). Four growth periods that included 0–45, 46–75, 76–95 and 96–105 days after germination were used to generate AUDPC. The AUDPC was calculated using the equation described in Section 2.6.1.

After the physiological maturity, the panicles from tagged plants were harvested manually, dried for 48 h then pooled to form a bulk sample from each plot, threshed and moisture content adjusted to 12.5% with the help of Pfeuffer HESO grain moisture meter. The grain weights (kg), weight of 100 seeds, grain yields and harvest index were determined. Harvest index was determined by calculating the ratio of harvested grain yield and biological yield (grain yield/biological yield).

Analysis of data recorded were performed by Statistical Analysis System software version 9.2 [41] and means separated by Least Significant Difference (LSD). Linear regression analysis was performed to determine the relationship between grain yield and anthracnose disease severity.

## 3. Results

### 3.1. Isolation and Pathogenicity Fungal Isolates

Following the isolation of fungal pathogens, seven *Colletotrichum* isolates were obtained. Pathogenicity was tested on sorghum detached leaves in vitro for the seven isolates. After 7 days of inoculation, all the tested isolates caused leaf necrosis with dark brown lesions, whereas controls had no symptoms. The typical symptoms produced by artificial inoculation conformed with the original symptoms under natural conditions. There was no significant difference in pathogenicity among the seven *Colletotrichum* isolates. Lesion diameters were not significantly different among isolates (Appendix A). The inoculated fungal isolates were re-isolated from the lesions on the inoculated leaves and no fungi were isolated from the control leaves. Thus, Koch’s postulates were fulfilled and the isolates of the seven *Colletotrichum* species were determined to be the pathogens causing anthracnose of sorghum. Koch’s postulate was verified by identifying re-isolated strains based on morphological characteristics and ITS sequences.

### 3.2. Morpho-Cultural Characterization of Colletotrichum spp. Isolates

Following isolations of fungal pathogens, seven *Colletotrichum* isolates were obtained and described based on the morphological and cultural characteristics (Table 3). The mycelial and colony characteristics of seven *Colletotrichum* isolates were analysed after 10 days of isolates growth on PDA medium. Variations in colony color of the *Colletotrichum* isolates on PDA were observed and the isolates were separated into three groups: Group I (consisting of two isolates namely KAT001-A and W008-L) was characterized by the white color on both upper and reverse side (Figure 1A), Group II (consisting of one isolate namely MUT006-B) was characterized by cottony white and orange on the upper and reverse, respectively, (Figure 1B) and Group III (consisting of four isolates namely KAT015-A, MTH022-A, MAK017-D and IKA009-G) was characterized by cottony white and yellow on the upper and reverse side, respectively (Figure 1C). There was no variation in conidia characteristics as all conidia were falcate, hyaline and smooth with no septate (Figure 1). The appressoria characteristics were similar for all the groups of the isolates, characterized by unicellular, black globose-shaped (Figure 1). Significant differences in growth rates were observed among the seven isolates on PDA medium. After 10 days of growth, the colony size varied from 5.44 cm to 8.50 cm on PDA (Table 3). The mean conidia length ranged from 23 ± 1.92 to 35.27 ± 2.14 µm and width from 4.47 ± 0.53 to 5.29 ± 0.81 µm (Table 3). These morphological characteristics confirmed the identity of the fungal pathogens to be *Colletotrichum sublineola*. *Colletotrichum sublineola* identified in this study is closely related *Colletotrichum eremochloae*. The differences between the two *Colletotrichum* species are presented in Appendix A.

### 3.3. PCR Amplification and DNA Sequencing

PCR amplification was performed using the *Colletotrichum* specific (COL) and ITS primers which successfully amplified products of approximately 560 bp and 500 bp, respectively.

Results of ITS DNA sequences of the seven *Colletotrichum* isolates, when compared with those of *Colletotrichum* accessions in the GenBank database, showed that all the seven isolates had 100% similarity to that of *Colletotrichum sublineola* (Table 4). Based on nucleotide BLAST search the similarity of the isolated fungi to the closest species available in the NCBI were 100% *Colletotrichum* species (Table 4). The nucleotide sequences of ITS region for seven isolates were submitted to NCBI database and were assigned the following accession numbers: ON764330 (KAT015-A), ON764342 (KT001-A), ON764382 (W008-L), ON764366 (MUT006-B), ON764351 (MAK017-D), ON764322 (IKA009-G) and ON764362 (MTH022-A) (Table 4).

### 3.4. Phylogenetic Analysis of ITS Region

The BLASTn searches showed that the ITS sequences of the seven isolates were 100% identical to *Colletotrichum sublineola*. For phylogenetic analysis, the neighbour-joining tree based on the datasets of ITS was constructed and includes the seven isolates from this study and 11 referenced *Colletotrichum* strains from the database (Appendix A). Phylogenetic analysis revealed that the seven isolates (KAT015-A, KT001-A, W008-L, MUT006-B, MAK017-D, IKA009-G and MTH022-A were clustered with the *Colletotrichum sublineola* clade (Figure 2), which was consistent with the homology search results that were conducted using BLASTn.

### 3.5. Reactions of Sorghum Genotypes to Colletotrichum Sublineola

The response of 14 sorghum genotypes was evaluated after inoculation with *Colletotrichum sublineola* in the greenhouse and under field conditions.

In the greenhouse experiments, significant (*p* ≤ 0.05) differences were observed among the sorghum genotypes infected with *C. sublineola* (Table 5). Three weeks after inoculation in the greenhouse, purple, red, circular to elongated elliptical necrotic spots and tan or black spots with acervuli were the common symptoms observed on sorghum leaves (Figure 3). These symptoms were observed in all the genotypes except for Mugeta, Kauwi, Marcia and KARI Mtama-1 (Table 5). Sorghum genotypes Kaguru, Kateng’u, Rasta and Makueni Local-2 recorded the highest anthracnose disease severity (Table 5). No anthracnose disease symptoms were observed from the control plants inoculated with sterile distilled water (Figure 3). After 12 weeks post-inoculation, necrotic lesions were collected, *C. sublineola* was re-isolated and was confirmed to be similar to the isolate used in inoculation.

The anthracnose response for the 14 sorghum genotypes under field conditions is presented in Table 6 and Table 7. Significant (*p* < 0.001) differences were observed for both AUDPC and anthracnose disease severity among the sorghum genotypes under field conditions during both experimental seasons (Table 6 and Table 7). Anthracnose symptoms were observed on all genotypes except for Mugeta, Kauwi, Marcia and KARI Mtama-1. Marcia, Kari Mtama-1, Kauwi and Mugeta showed a resistance to anthracnose disease with a severity score of 1 (no necrotic lesions observed on the leaves and sheath) at both seasons of KALRO-Ithookwe and KALRO-Kiboko fields. The mean severity of anthracnose disease at physiological maturity was 3.78 at Ithookwe and 3.46 at Kiboko (Table 6). Rasta shows high susceptibility by scoring a severity score of 5. Kaguru, used as susceptible (negative) control, scored 4.96 at Ithookwe and 4.84 in Kiboko while Kateng’u scored 4.94 Ithookwe and 4.84 at Kiboko (Table 6). Serena, Gadam, IESV24029SH, Kiboko local 2 and Makueni local 1 showed moderate susceptibility by recording a disease severity that ranged between 4.0 and 4.9 (Table 6).

Sorghum genotypes were classified into different disease severity classes based on the disease rating scale, which was the final anthracnose severity recorded at 12 weeks post-inoculation (Table 6 and Table 7). The genotypes showed a similar infection response in both seasons for both KALRO-Kiboko and KALRO-Ithookwe field stations with four genotypes rated as resistant (R) accounting for 28.57% of the genotypes, seven genotypes rated as moderately susceptible (MR) accounting for 50% of the total and three genotypes rated as susceptible (S) accounting for 21.42% of the total. There were no genotypes moderately resistant and highly susceptible.

### 3.6. Grain Yield, 100 Seed Weight and Harvest Index

Significant (*p* ≤ 0.05) differences were observed in grain weight, grain yield, weight of 100 seeds and harvest index among the tested sorghum genotypes (Table 8). Makueni Local-2 recorded the highest weight of 54.35 g while Mugeta (16.74 g) recorded the lowest (Table 8). The genotype with highest grain yield was KARI Mtama-1 with 3.65 t/ha with Mugeta recording the lowest with 0.35 t/ha. Kiboko Local-2 and Makueni Local-2 had seeds with highest weight (2.8 g) when 100 seeds were measured while Kateng’u (1.6 g) had the seeds with lightest weight (Table 8). KARI Mtama-1 and Marcia had the highest harvest index of 0.41 while Mugeta had the lowest harvest index of 0.06 (Table 8).

There were significant (*p* < 0.001) differences in grain weight, grain yield, 100 seeds and harvest index among sorghum genotypes at both KARLO Kiboko and Ithookwe field stations (Table 8). The grain weight ranged from 15.7 g to 53.9 g at KALRO Ithookwe, where Kiboko Local-2 and Mugeta recorded the highest and lowest weights, respectively. At KALRO Kiboko, the highest and lowest grain weight was recorded for sorghum genotypes Makueni Local-1 (61 g) and Kateng’u (14.4 g), respectively (Table 8).

Grain yield at KALRO Ithookwe ranged from 0.5 t/ha to 3.7 t/ha while at KALRO Kiboko site ranged from 0.2 t/ha to 3.6 t/ha. The genotypes KARI Mtama-1 and Mugeta recorded the highest and lowest grain yields, respectively, at both field sites. The weight of 100 seeds for the tested genotypes was recorded and ranged from 1.5 g to 2.7 gat Ithookwe where Kiboko Local-2 and Kateng’u recorded the highest and lowest weights. At KALRO- Kiboko the genotypes Makueni Local-2 (3.1 g) and Kateng’u (1.7 g) recorded the highest and least 100 seed weights, respectively (Table 8). Among all the genotypes, KARI Mtama-1 and Marcia recorded the highest harvest index (0.4) while Mugeta, Rasta and Kaguru had the lowest harvest index that ranged from 0.03 to 0.2 (Table 8).
$$\mathrm{Grain}\;\mathrm{yield}=\frac{\mathrm{grain}\;\mathrm{weight}\;\left(g\right)}{100A}$$
where A = (Row harvested × Interval between rows × Length of the plot)m^2^. $ Harvest index = grain yield/biological yield.

### 3.7. Regression Analysis on Effect of Anthracnose on Grain Weight, Grain Yield and Weight of 100 Seeds

The linear regression analysis results showed no significant (*p* > 0.005) effect of sorghum anthracnose disease on grain weight, grain yield and weight of 100 seeds (Appendix A and Appendix A).

## 4. Discussion

Anthracnose is one of the most devastating diseases of sorghum due to the seed-borne nature of the pathogen. In lower eastern Kenya, anthracnose is one of the main constraints for sorghum crop production. However, to date, no study has focused on the identification and characterization of the pathogen, which is fundamental to understand the scope of the disease in the region. The correct identification of the fungal species and pathogenicity can provide important support for disease management and control as well as identifying possible resistance sources for exploitation in sorghum breeding programs. In the present study, seven isolates obtained from the leaves of sorghum plants showing anthracnose symptoms in farmers’ fields in lower eastern Kenya, were identified as *Colletotrichum* species based on morphology. The seven isolates were identified to species level based on a combination of morphological characters and ITS sequence data. In this study, we evaluated the pathogenicity of the seven isolates using a rapid, stable and efficient wounding inoculation method. Unlike the non-wounding method, this method involves in vitro inoculation and incubation under controlled laboratory conditions, has the advantages of speed, accuracy (which has fewer environmental effects) and efficiency (ability to conduct multiple, repeated large-scale evaluations) and will not cause pathogen accumulation in the field.

This is the first report characterizing *Colletotrichum* species associated with anthracnose of sorghum in lower eastern Kenya using morphological and phylogenetic analysis based on ITS sequence. In the past, investigating the aetiology of sorghum anthracnose in Kenya has been largely ignored due to the historically minor impact of the disease [10]. This current study was performed with the aim of morphological and molecular identification of *Colletotrichum* species associated with sorghum anthracnose as well as the impact of the disease on yields. Spore morphology, colony characteristics and ITS gene sequences comparisons identified *C. sublineola*, responsible for sorghum anthracnose in Kenya. In this study, the cultural and morphological characteristics of the pathogen were identical to the findings of Choi et al. [46]. Variations in mycelia growth observed were similar to the study by Gabriela et al. [47], who reported a significant variation in mycelia growth of *Colletotrichum* species. The morphological descriptions of the seven isolates in this study were in agreement with previous descriptions of *Colletotrichum* species, although the cultural characteristics may be diverse due to different cultural conditions, temperature, light regime and geographic isolates, and therefore conidia morphology alone could not distinguish all species with species complexes [48,49,50].

The study of *Colletotrichum* species is remarkably difficult because of its complexity and diversity. Morphological characteristics alone cannot identify *Colletotrichum* species [51,52] due to morphological uniformity within the genus and the fact that there are few distinctive morphological characters available for identification [18]. As morpho-cultural characterization alone is not reliable criteria for identification of *Colletotrichum* species [53], we used the ITS sequences to achieve a more accurate identification. In our study based on the ITS-gene phylogenetic analysis, the seven isolates were identified as *C. sublineola*. The molecular phylogenetic evidence showed species level divergence between *C. sublineola* identified in this study and *Colletotrichum eremochloae*. Identifying the causal agent is the first and most significant step to facilitate resistance breeding programs and to determine effective control strategies for disease management programs.

Anthracnose is considered the most destructive diseases of sorghum because it infects all aerial parts of the plant. The use of genetic resistance is the most efficient, cost-effective and environmental-friendly strategy of combating the damage caused by anthracnose disease of sorghum [22]. Therefore, it is important to evaluate the response of different sorghum genotypes to *C. sublineola* in a particular region to provide genetic resources for sorghum breeding programs. Results from previous evaluations of sorghum germplasm accessions from different countries including Burkina Faso, Ethiopia, Mali, Mozambique, Sudan, South Africa, Uganda, Zimbabwe and China have identified resistant germplasm to anthracnose disease [24,25,26,27,28,29]. Nevertheless, the identification of additional sources of resistance from many diverse regions is required to adequately control the disease worldwide and therefore continuous efforts are needed to find sorghum germplasm resistant to the disease. The present study revealed marked variations in anthracnose disease severity and area under the disease progress curve among the tested sorghum genotypes in both greenhouse and field conditions. About 28.57% of the 14 genotypes exhibited a resistance response. The infection response in the greenhouse and both seasons in KALRO Kiboko and Ithookwe field sites were similar, which did not affect the resistant reaction classes. Variable reaction of sorghum genotypes to anthracnose under natural infection in Ethiopia has also been reported by Chala et al. [7], and the resistance reaction is significantly affected by the genetic background of the genotypes [29].

The results from the current study showed differential reactions of sorghum genotypes to anthracnose under greenhouse and natural conditions. The 14 tested sorghum genotypes exhibited resistance, susceptible and highly susceptible to anthracnose in both greenhouse and two seasons of field trials in the two sites, indicating that Kenya sorghum accessions are important sources of anthracnose resistance. The diversity of sorghum genotypes responding to anthracnose would suggest the existence of genetic variation for host resistance. Similarly, the existence of resistant and susceptible response of sorghum germplasm to anthracnose has reported at Texas, USA in two growing seasons [29]. Cuevas et al. [24] and Cuevas et al. [25] evaluated sorghum accessions from Zimbabwe, Burkina Faso and South Africa and reported variable responses of sorghum accessions to anthracnose. The differences in reaction responses of sorghum genotypes to anthracnose indicated the inherent genetic resistance which can be exploited by breeders in breeding programs. The identification of highly resistant and stable genotypes offers the possibility of deploying these resistances in sorghum breeding programmes and future opportunities to deliver resistant cultivars in target regions for the effective control of the disease.

Sorghum genotypes Rasta, Kaguru and Kateng’u were highly susceptible and produced low grain weight and grain yield. This is in agreement with reports by Prom et al. [54], who indicated that anthracnose disease significantly reduces yield in susceptible sorghum accessions. It is known that anthracnose has a strong negative impact on grain-filling and this result in a large amount of yield losses on susceptible genotypes. In addition, the conidia of *C. sublineolum*, when they are inside the plant, interfere with water and nutrient movement in the vascular tissues resulting in poor development of the panicle and grain [55]. Foliar anthracnose causes damages to the leaves and stalk rot reducing the photosynthetic area which results in a reduction in grain sizes, grain weight and subsequent yield losses [56]. Despite high disease severity recorded on sorghum genotypes Makueni Local-2, Kiboko Local-2 and IESV24029SH recorded the highest grain weight and grain yield. This is in agreement with the report by Thakur et al. [40] where the effect of foliar anthracnose severity on grain yield and related traits are dependent on the genotype and environment. Therefore, the effect of anthracnose on yield may also be affected by many factors that influence the disease epidemic. The development of anthracnose disease epidemic is affected by the availability of optimum temperature, relative humidity, host tissue, level of host resistance, *Colletotrichum* pathotypes and other factors during the growing periods of the crop [48,55]. Therefore, field trials were carried out to verify the results of greenhouse observations. The results were similar to the resistance rankings of sorghum genotypes screened in the greenhouse.

## 5. Conclusions

The *Colletotrichum* species isolates collected from sorghum fields in eastern Kenya had a variation in cultural and morphological characteristics when grown on potato dextrose agar. Based on ITS sequences and phylogenetic analysis, the seven *Colletotrichum* species causing sorghum anthracnose were identified as *C. sublineola*. There is a need to carry out further molecular characterization of the identified *Colletotrichum sublineola* using multi-gene sequence data. Fourteen sorghum genotypes screened for resistance to anthracnose disease, showed differential reactions to anthracnose under greenhouse and natural conditions. The observed genetic differences among genotypes suggest the existence of heritable variations, which could be exploited in sorghum breeding programs for the development of varieties possessing resistance to anthracnose disease.

## Figures and Tables

**Figure 1 jof-09-00100-f001:**
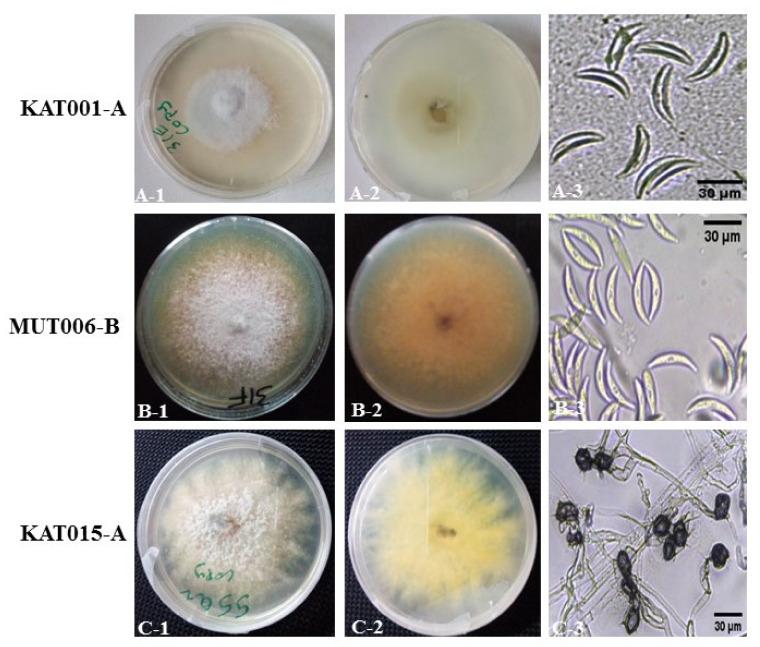
Morphological variability in the colonies of *Colletotrichum* isolates associated with sorghum anthracnose. KAT001-A, MUT006-B and KAT015-A are the representative isolates of the three groups. Each row from left to right: views of the top and bottom of potato dextrose agar (PDA) plate of representative isolates and conidia. (**A-1**,**A-2**) (KAT001-A) white both at top and reverse view; (**B-1**,**B-2**), (MUT006-B) white top view and salmon orange on the reverse view and (**C-1**,**C-2**), (KAT015-A) white and cottony on the upper view and yellow on reverse view with filamentous mycelial growth. (**A-3**), Falcate, smooth and hyaline conidia; (**B-3**), Hyaline, falcate, smooth conidia with nuclear spot at the center and (**C-3**), Black, globose shaped, single celled primary appressoria.

**Figure 2 jof-09-00100-f002:**
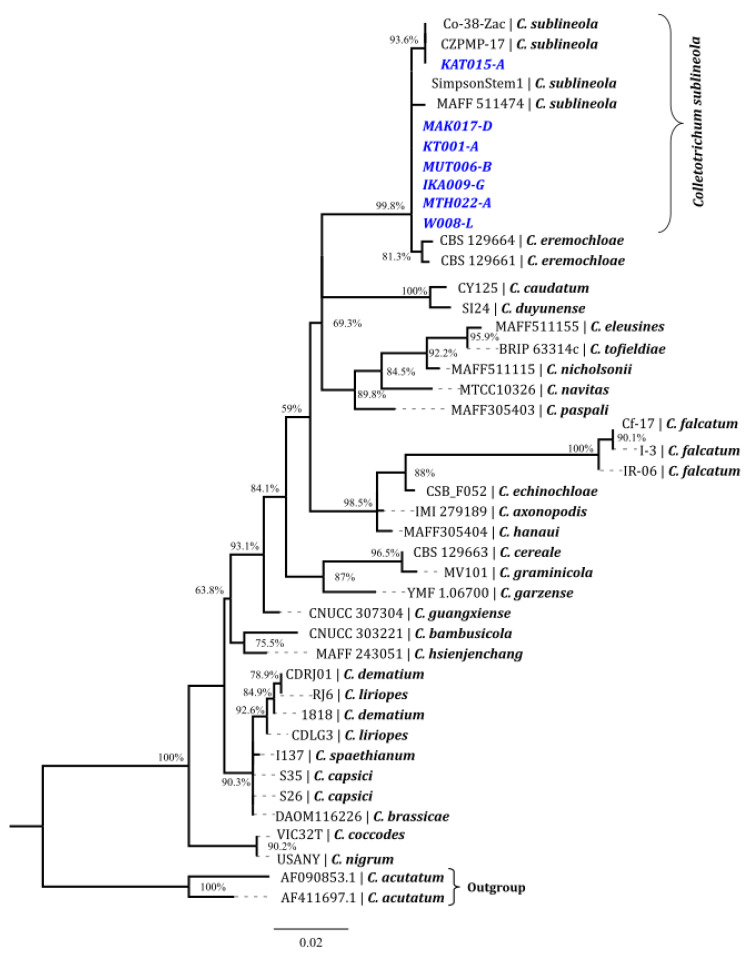
Phylogenetic analysis of *Colletotrichum sublineola* constructed with the sequences for ITS regions. Phylogenetic relationships were inferred using the neighbor-joining method. The values at the nodes represent bootstrap proportions from 1000 bootstraps. The evolutionary distances were calculated using Kimura-2-parametre with different gamma shape parameters for each tree. All the positions containing gaps were deleted. The tree was rooted to the two members of *Colletotrichum acutatum* and was drawn to scale.

**Figure 3 jof-09-00100-f003:**
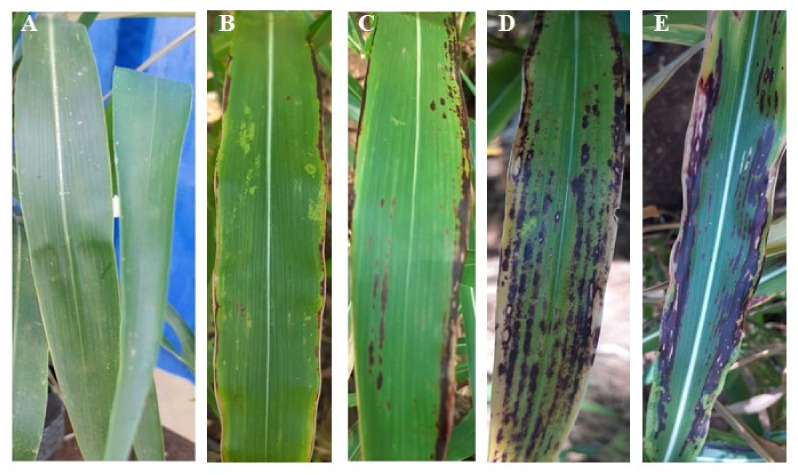
Reaction of sorghum genotypes to *Colletotrichumsublineolum* inoculation under greenhouse conditions. (**A**) represents the non-inoculated control, (**B**) moderately resistant with presence of yellow flecks with 1–10% of the leaves having necrotic lesion without acervulus, (**C**) moderately susceptible with 1–30% of leaves having necrotic lesions and few acervulus and (**D**,**E**) susceptible with 1–50% of leaves having necrotic lesions and more acervuli present.

**Table 2 jof-09-00100-t002:** Assessment scale of anthracnose disease against sorghum plants under field conditions.

Severity Ratings	Symptoms and Lesion Type on the Leaves	DRC
1	No visible symptoms/chlorotic flecks	R
2	1–25% leaf area covered with small, restricted lesions	MR
3	26−50% leaf area covered with small, restricted lesions	MS
4	51−75% leaf area covered with large, coalescing lesions	S
5	>75% leaf area covered with large, coalescing lesions	HS

The assessment scale is adopted from Thakur et al. [40]. DRC—Disease Reaction Classes: Resistant plants (R) were rated as 1, moderately resistant plants (MR) as 2, moderately susceptible (MS) as 3, susceptible plants (S) as 4 and highly susceptible plants (HS) as 5.

**Table 3 jof-09-00100-t003:** Morpho-cultural characteristics of *Colletotrichum* isolates in lower eastern Kenya.

Isolate ID Code	County	Location	Mycelial Characteristics	Colony Size (cm)	Colony Growth Rate/Day (cm)	Conidia		Macroscopic Characteristics	Microscopic Characteristics	Pathogen Identity
						Length (µm) (Average ± SD *)	Width (µm) (Average ± SD *)			
KAT015-A	Makueni	Kathonzweni	Raised	7.46 b	0.82 c	34.47 ± 2.14 a	5.12 ± 1.10 a	Top—White Reverse—Yellowish	Falcate, No septation and hyaline	*Colletotrichum sublineola*
KT001-A	Kitui	Kitui township	Cottony	8.5 ab	0.98 a	32.47 ± 2.14 a	4.59 ± 1.10 b	Top—Cotton Reverse—Whitish centers	Falcate, hyaline with no septation	*Colletotrichum sublineola*
W008-L	Makueni	Wote	Cottony	7.6 b	0.9 b	31.41 ± 5.37 a	4.47 ± 0.53 b	Top—Cotton Reverse—Whitish with dark centers	Falcate, hyaline with no septation	*Colletotrichum sublineola*
MUT006-B	Kitui	Mutomo	Raised and cottony	7.69 b	0.88 bc	35.27 ± 2.14 a	5.02 ± 1.14 ab	Top—white Reverse-pale orange	Falcate, slender, hyaline and no septation	*Colletotrichum sublineola*
MAK017-D	Makueni	Makindu	Raised and cottony	6.03 bc	0.74 cd	34.47 ± 2.14 a	5.12 ± 1.10 a	Top—White Reverse—Yellowish	Falcate, No septation and hyaline	*Colletotrichum sublineola*
IKA009-G	Kitui	Ikanga	Raised and cottony	7.39 b	0.88 bc	23 ± 1.92 b	4.91 ± 0.51 ab	Top—White Reverse—Yellowish	Falcate, No septation and hyaline	*Colletotrichum sublineola*
MTH022-A	Machakos	Muthesya	Raised and cottony	5.44 c	0.54 d	31.41 ± 5.37 a	5.29 ± 0.81 a	Top—White Reverse—Yellowish	Falcate, No septation and hyaline	*Colletotrichum sublineola*

* SD means standard deviation. Values within the same column followed by different small letters mean that they are significantly different based on variance with the least significant difference test at *p* = 0.05.

**Table 4 jof-09-00100-t004:** Similarity of ITS sequences of *Colletotrichum* species isolated from sorghum plants with anthracnose disease in Kenya, compared with that of the accessions in the GenBank database.

Isolate ID Code	County	ITS Accession	Closest Match in Blast	Size (bp)	Similarity (%)
KAT015-A	Makueni	ON764330	*Colletotrichum sublineola*	567	100
KT001-A	Kitui	ON764342	*Colletotrichum sublineola*	538	100
W008-L	Makueni	ON764382	*Colletotrichum sublineola*	549	100
MUT006-B	Kitui	ON764366	*Colletotrichum sublineola*	564	100
MAK017-D	Makueni	ON764351	*Colletotrichum sublineola*	563	100
IKA009-G	Kitui	ON764322	*Colletotrichum sublineola*	564	100
MTH022-A	Machakos	ON764362	*Colletotrichum sublineola*	516	100

**Table 5 jof-09-00100-t005:** Area under the disease progress curve (AUDPC), disease severity scores and disease reaction types in 14 sorghum genotypes following inoculation with *Colletotrichum sublineolum* (Isolate KT001-A) under greenhouse conditions.

Genotype	AUDPC	Disease Severity Score	DRC
Dark Red	5.57 bc	2.78 bc	MR
Gadam	5.94 ab	2.97 bc	MR
IESV 24,029 SH	5.17 c	2.58 c	MR
Kaguru	7 a	3.5 b	MS
KARI Mtama-1	2 e	1 e	R
Kateng’u	6.7 ab	3.35 b	MS
Kauwi	3.8 d	1.9 d	R
Kiboko Local-2	5.1 c	2.55 c	MR
Makueni Local-2	6.52 ab	3.27 bc	MS
Marcia	2 e	1 e	R
Mugeta	2 e	1 e	R
Rasta	7.67 a	3.83 ab	MS
Seredo	5.77 b	2.88 bc	MR
Serena	5.47 bc	2.73 c	MR

AUDPC: Calculated using disease severity data recorded for four-time period (1–3, 3–6, 6–9, 9–12 weeks after inoculation) using formula described by Madden et al. [42]. Disease severity obtained by calculating the mean severity scores of each sorghum cultivars recorded for a twelve-week period; values in disease severity column are means of each sorghum variety obtained by averaging total means per replication for data recorded for twelve weeks (n = five plants per replication with three replications). Ranking determined by Turkey’s for post-hoc analysis in SAS software, values with similar letters in the same column are not significantly different at *p* = 0.05 according to Fisher’s least significance test; DRC—disease reaction classes. Resistant plants (R), moderately resistant plants (MR, moderately susceptible (MS), susceptible plants (S) and highly susceptible plants (HS).

**Table 6 jof-09-00100-t006:** Area under the disease progress curve (AUDPC), disease severity scores and disease reaction types in 14 sorghum genotypes following inoculation with *Colletotrichum sublineola* under field conditions at KALRO-Kiboko.

	Kiboko Season 1			Kiboko Season 2		
	AUDPC	Severity	DRC	AUDPC	Severity	DRC
Dark Red	35.16 e	2.38 c	MR	51.03 d	3.31 bc	MS
Gadam	50.46 de	3.07 b	MS	59.71 c	3.49 bc	MS
IESV 24,029 SH	47.33 de	3.09 b	MS	54.75 cd	3.3 c	MS
Kaguru	57.8 b	3.64 a	S	76.83 a	4.34 a	S
KARI Mtama-1	20 ef	1 d	R	20 f	1 f	R
Kateng’u	54.7 d	3.42 a	MS	75.29 a	4.29 a	S
Kauwi	12.94 f	1.14 d	R	25.75 e	1.48 d	R
Kiboko Local-2	54.41 d	3.26 ab	MS	63.97 b	3.74 b	S
Makueni Local-2	55.66 d	3.5 a	S	64.42 b	3.83 ab	S
Marcia	20 ef	1 d	R	20 f	1 f	R
Mugeta	20.79 ef	1.09 d	R	22.42 f	1.2 e	R
Rasta	64.27 a	3.81 a	S	78.46 a	4.39 a	S
Seredo	33.5 e	2.33 c	MR	47 de	3.12 c	MS
Serena	51.19 de	3.08 b	MS	57.92 c	3.44 bc	MS

AUDPC: Calculated using disease severity data recorded for four-time phonological growth period (0–45, 46–75, 76–95, 96–105 days after planting) using formula described by Madden et al. [42]. Disease severity obtained by calculating the mean severity scores of each sorghum cultivars recorded at 45th, 75th, 95th and 105th growth period in each season. Values in the table are means of each sorghum varieties per replication; values in disease severity column are means of each sorghum variety obtained by averaging total means per replication for data recorded at 45th, 75th, 95th and 105th day after sowing (n= 10 plants per replication with four replications). Ranking was determined by Turkey’s for post-hoc analysis in SAS software, values with similar letters in the same column are not significantly different at *p* = 0.05 according to Fisher’s least significance test. DRC—disease reaction classes. Resistant plants (R), moderately resistant plants (MR, moderately susceptible (MS), susceptible plants (S) and highly susceptible plants (HS).

**Table 7 jof-09-00100-t007:** Area under the disease progress curve (AUDPC), disease severity scores and disease reaction types in 14 sorghum genotypes following inoculation with *Colletotrichum sublineolum* under field conditions at KALRO-Ithookwe.

	Ithookwe Season 1			Ithookwe Season 2		
	AUDPC	Severity	DRC	AUDPC	Severity	DRC
Dark Red	34.89 d	2.7 c	MS	44.98 b	3.01 b	MS
Gadam	74.11 a	4.1 a	S	76.01 a	4.3 a	S
IESV 24,029 SH	52.21 b	3.49 b	MS	46.5 b	3.11 b	MS
Kaguru	43.18 bc	3.1 bc	MS	70 a	4.37 a	S
KARI Mtama 1	20 e	1 d	R	20 d	1 d	R
Kateng’u	53.83 b	3.61 b	S	75.64 a	4.4 a	MS
Kauwi	22.33 e	1.27 d	R	26.63 c	1.55 c	R
Kiboko Local-2	40.42 c	2.97 bc	MS	46.83 b	3.06 b	MS
Makueni Local-2	36.61 d	2.79 c	MS	48.67 b	3.21 b	MS
Marcia	20 e	1 d	R	20 d	1 d	R
Mugeta	21.08 e	1.14 d	R	22.02 d	1.17 d	R
Rasta	43.02 bc	3.11 bb	MS	66.75 a	4.17 a	S
Seredo	35.52 d	2.73 c	MS	46.63 b	3.07 b	MS
Serena	34.9 d	2.7 c	MS	47.58 b	3.1 b	MS

AUDPC: Calculated using disease severity data recorded for four-time phonological growth period (0–45, 46–75, 76–95, 96–105 days after planting) using formula described by Madden et al. [42]. Disease severity obtained by calculating the mean severity scores of each sorghum cultivars recorded at 45th, 75th, 95th and 105th growth period in each season. Values in the table are means of each sorghum varieties per replication; values in disease severity column are means of each sorghum variety obtained by averaging total means per replication for data recorded at 45th, 75th, 95th and 105th day after sowing (n = 10 plants per replication with four replications). Ranking was determined by Turkey’s for post-hoc analysis in SAS software, values with similar letters in the same column are not significantly different at *p* value 0.05 according to Fisher’s least significance test; DRC—disease reaction classes. Resistant plants (R), moderately resistant plants (MR, moderately susceptible (MS), susceptible plants (S) and highly susceptible plants (HS).

**Table 8 jof-09-00100-t008:** Grain weight, grain yield, 100 seed weight and harvest index of sorghum genotypes at KALRO Ithookwe and Kiboko field sites for two seasons.

	KALRO-Ithookwe-2021 (Season 1 & 2)				KALRO-Kiboko-2021 (Season 1 & 2)			
	* Grain Weight (g)	# Grain Yield (t/ha)	100 Seeds Weight (g)	$ Harvest Index	* Grain Weight (g)	# Grain Yield (t/ha)	100 Seeds Weight (g)	$ Harvest Index
Gadam	35.7 d	1.8 h	2.2 c	0.31 b	32.2 ef	1.6 f	2.2 d	0.26 c
Kateng’u	20.6 ef	1.32 hi	1.5 d	0.26 d	14.4 h	1.5 f	1.7 e	0.28 c
Marcia	46.9 c	3.2 c	2.4 b	0.4 a	54.5 b	3.4 b	2.7 ab	0.41 a
IESU24029 SH	43.5 c	2.9 d	2.4 b	0.26 d	42.6 c	2.3 e	2.4 c	0.26 c
Kauwi	43.2 c	2.6 e	2.4 b	0.31 b	35.2 e	2.1 ef	2.5 b	0.3 b
KARI Mtama-1	51.4 b	3.7 a	2.6 a	0.41 a	55.4 b	3.6 a	2.9 a	0.4 a
Kiboko Local-2	53.9 a	3.3 c	2.7 a	0.26 d	44.34 c	3.4 b	2.9 a	0.29 b
Rasta	16.3 f	0.8 i	1.8 cd	0.15 e	17.7 g	1.2 g	1.9 e	0.21 d
Makueni Local-2	47.7 c	2.9 d	2.5 a	0.28 c	61 a	3.1 c	3.1 a	0.29 b
Serena	48.6 c	2.3 f	2.5 a	0.32 b	40.5 d	2 ef	2.3 d	0.28 b
Mugeta	15.7 f	0.5 j	2.2 c	0.08 f	17.8 g	0.2 h	2.2 d	0.03 e
Seredo	36.9 d	2 g	1.9 cd	0.33 b	37.9 e	2 ef	2.1 de	0.3 b
Kaguru	28.4 e	0.9 i	1.8 cd	0.2 de	25.1 f	0.75 gh	2 de	0.15 de
Dark red	29 e	1.5 hi	1.9 cd	0.2 de	30.4 ef	1.5 f	2 de	0.24 c
Mean	36.99	1.89	2.2	0.27	36	2.1	2.4	0.3

Ranking was determined by Turkey’s for post-hoc analysis in SAS software, values with similar letters in the same column are not significantly different at *p*-value 0.05 according to Fisher’s least significance test; values in the columns are means of each sorghum varieties per replication for data recorded at 45th, 75th, 95th and 105th day after sowing (n = 10 plants per replication with four replications). * Grain weight was determined by calculating the mean weight in grams of grains harvested in 4 inner rows per plot. # Grain yield was calculated according to the following formula.

## Data Availability

The nucleotide sequences of ITS region for seven isolates obtained in this study were deposited in Genebank through an online submission portal and were assigned the following accession numbers: ON764330, ON764342, ON764782, ON764366, ON764351, ON764322 and ON764362.

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
