# Peer review of "Identification and Characterization of Colletotrichum Species Causing Sorghum Anthracnose in Kenya and Screening of Sorghum Germplasm for Resistance to Anthracnose"

_jof, 2023, doi:10.3390/jof9010100_

Round 1
Reviewer 1 Report
1. How many samples were obtained from each county?
2. How many leaves were investigated in greenhouse screening trials and field screening trials?
3. ‘Table 2. Assessment scale of anthracnose disease against sorghum plants under greenhouse conditions. ’ is right? Greenhouse?
4. Statistical analysis of the data in Table 3 should be made.
5. Only ITS primers were used for identification of Colletotrichum species. Why not use other gene primers to identify the species?
6. How the data in Tables 5 and 6 were obtained?
7. ‘Linear regression analysis was performed to determine the relationship between grain yield and anthracnose disease severity.’, how about the results?
8. What is harvest index? How to obtain harvest index?
9. In ‘(p ≤0.05)’ and ‘(P<0.001)’, p and P should be in consistent format.
Author Response
22nd December, 2022
The Editor-in-Chief
MDPI Journal of Fungi
Dear Madam,
RE: Submission of Revised Manuscript: microbiolres-2060908
It is with pleasure that we submit to you the revised version of manuscript Ref. microbilres-2060908 titled “Identification and Characterization of Colletotrichum Species Causing Sorghum Anthracnose in Kenya and Screening of Sorghum Germplasm for Resistance to Anthracnose” for publication in MDPI Journal of Fungi. The authors appreciate the time and effort by the editor and reviewers for reviewing this manuscript and for providing insightful comments and suggestions to improve its quality. We have revised the manuscript as per the reviewers’ suggestions and comments. We have incorporated the following reviewer’s specific comments in the revised version of manuscript. The point-by-point responses are provided below.
Response to Comments from Reviewers
Reviewer #1
Comments and Suggestions for Authors
- How many samples were obtained from each county?
Response: Thank you for the observation. We have revised and included the number of samples used from each of the three Counties.
- How many leaves were investigated in greenhouse screening trials and field screening trials?
Response: Thank you for the feedback. We have provided the information in the methods section.
- ‘Table 2. Assessment scale of anthracnose disease against sorghum plants under greenhouse conditions. ’ is right? Greenhouse?
Response: Thank you for the comment. The assessment scale of anthracnose disease under greenhouse conditions is correct.
- Statistical analysis of the data in Table 3 should be made.
Response: We appreciate suggestion from the reviewer, and we have made statistical analysis of the data in Table 3.
- Only ITS primers were used for identification of Colletotrichum Why not use other gene primers to identify the species?
Response: Thanks for pointing out the use of other gene primers to identify the species. Although phylogenetic analysis based on ITS of the isolates showed that the isolates belong to the same species and were clearly separated in the dendrogram, we have recommended further molecular identification and characterization of Colletotrichum sublineola using multi-gene sequence data.
- How the data in Tables 5 and 6 were obtained?
Response: Thank you for the feedback. In our materials and methods section 2.6.2, we have provided information on how data in Tables 5 and 6 was obtained. In addition, results section, we have provided foot notes for Tables 5 and 6, on how the data was collected.
- ‘Linear regression analysis was performed to determine the relationship between grain yield and anthracnose disease severity. How about the results?
Response: We appreciate the feedback. In our results section 3.7, we have provided results on the relationship between grain yield and anthracnose disease severity under the supplementary Table 5 and Supplementary Figure 1.
- What is harvest index? How to obtain harvest index?
Response: We appreciate the feedback. In our materials and methods section 2.6.2, we have added information how harvest index was obtained “Harvest index was determined by calculating the ratio of harvested grain yield and biological yield (grain yield/biological yield)”.
- In ‘(p ≤0.05)’ and ‘(P<0.001)’, p and P should be in consistent format.
Response: Thank you for the observation. We have revised the whole document and used “p“for consistency.
Reviewer 2 Report
This article describes the identification and characterization of Colletotrichum species causing sorghum anthracnose in Kenya and screening of resistant sorghum germplasms. The authors isolated 7 Colletotrichum strains from diseased sorghum in kenya. Those strains were fulfilled with Koch's postulates and identified as C. sublineola based on morphology observations, ITS sequence blasting results and single ITS gene phylogenetic analysis. The disease severity of 14 sorghum genotypes were evaluated by inoculation with C. sublineola strains in the greenhouse and disease surveys under field condition. The result reveals some sorghum genotypes are more resistance than others and could be used for sorghm breeding in the future. Generally,the paper is well-written and meets the scope of Journal of Fungi. However, in the part of phylogenetic analysis required more improvements(Fig. 2). Please check my comments about your manuscript below:
Major comments:
Figure 2 - (1) The closed related strain Colletotrichum eremochloa should be included in the phylogenetic analysis. (2) Please label strain number(ex. CBS131301) but not accession number of each strains used in the phylogram. (3) Authors may also provide basic strain information in a new table.
Table 3 - The pathogen identity in Table 3 should be Colletotrichum sublineola which have identified in this article. I also suggest to add strain characteristics of C. sublineola and the closed related strain C. eremochloa for comparison and discuss the difference.
Table 5,6,7,8 - Authors should briefly describe (1) statistical analysis method used and post hoc tests(if have) for analysis (2) n value (3) definition of significant (4) The values shown in table are mean or something else should be described clearly.
L26, L193, L371, L363, L498, The term "concatenated" in phylogenetics analysis means combine multiple genes, but in this article only use single ITS gene, please express more clearly.
L216-L218, The strain number used for inoculation should be mentioned.
L289-292, The author only describe no difference in pathogenicity among 7 Colletotrichum strains without any shown data. Please provide the result and describe the method used to estimate disease severity.
L293-295, How to prove the re-isolated fungi was the same as the inoculated one? Please describe.
L312-313, The growth rate of different strains significant varied, please explain the possible reasons and how it will affects pathogenicity test?
L348, The NCBI accession number ON764782 is invalid, please update.
- the initial pathogenicity test were use wound/drop inoculation method, but in the greenhouse screening trials were use spray method. The author should discuss more about the difference and possible impact on the outcome between wound and spray inoculation in the discussion section.
Minor comments:
L136 – no humidity shown.
L139 – spore suspension concentration should be use conidia/mL.
L140 – The authors should explain why the control is using PDA but not sterile distilled water.
L145, L165 - the degrees Celsius should after plus-minus sign.
Author Response
It is with pleasure that we submit to you the revised version of manuscript Ref. microbilres-2060908 titled “Identification and Characterization of Colletotrichum Species Causing Sorghum Anthracnose in Kenya and Screening of Sorghum Germplasm for Resistance to Anthracnose” for publication in MDPI Journal of Fungi. The authors appreciate the time and effort by the editor and reviewers for reviewing this manuscript and for providing insightful comments and suggestions to improve its quality. We have revised the manuscript as per the reviewers’ suggestions and comments. We have incorporated the following reviewer’s specific comments in the revised version of manuscript. The point-by-point responses are provided below.
Reviewer 2
Comments and Suggestions for Authors
This article describes the identification and characterization of Colletotrichum species causing sorghum anthracnose in Kenya and screening of resistant sorghum germplasm. The authors isolated 7 Colletotrichum strains from diseased sorghum in Kenya. Those strains were fulfilled with Koch's postulates and identified as C. sublineola based on morphology observations, ITS sequence blasting results and single ITS gene phylogenetic analysis. The disease severity of 14 sorghum genotypes was evaluated by inoculation with C. sublineola strains in the greenhouse and disease surveys under field condition. The result reveals some sorghum genotypes are more resistance than others and could be used for sorghum breeding in the future. Generally, the paper is well-written and meets the scope of Journal of Fungi. However, in the part of phylogenetic analysis required more improvements (Fig. 2). Please check my comments about your manuscript below:
Major comments:
Figure 2 - (1) The closed related strain Colletotrichum eremochloa should be included in the phylogenetic analysis. (2) Please label strain number (ex. CBS131301) but not accession number of each strains used in the phylogram. (3) Authors may also provide basic strain information in a new table.
Response: We appreciate the feedback. We have Colletotrichum eremochloa in the phylogenetic analysis. We have revised the labelling of the phylogram and used strain number instead of accession numbers. We have also provided the basic strain information as a supplementary Table.
Table 3 - The pathogen identity in Table 3 should be Colletotrichum sublineola which have identified in this article. I also suggest to add strain characteristics of C. sublineola and the closed related strain C. eremochloa for comparison and discuss the difference.
Response: Thank you the feedback. We have revised the pathogen identity in the table to Colletotrichum sublineola. We have also provided a table for the strain characteristics of C. sublineola and the closed related strain C. eremochloa as supplementary table and also discussed the differences.
Table 5,6,7,8 - Authors should briefly describe (1) statistical analysis method used and post hoc tests(if have) for analysis (2) n value (3) definition of significant (4) The values shown in table are mean or something else should be described clearly.
Response: Thank you the feedback. We have revised and provided the statistical analysis method and all the other details under the foot notes of each of the tables.
L26, L193, L371, L363, L498, The term "concatenated" in phylogenetics analysis means combine multiple genes, but in this article only use single ITS gene, please express more clearly.
Response: Thank you for the observation. We have revised and deleted the term "concatenated" since it is misplaced.
L216-L218, The strain number used for inoculation should be mentioned.
Response: Thank you for the feedback. We have included the strain number (Isolate KT001-A) used for inoculation of sorghum cultivars under greenhouse conditions.
L289-292, The author only describe no difference in pathogenicity among 7 Colletotrichum strains without any shown data. Please provide the result and describe the method used to estimate disease severity.
Response: Thank you for the feedback.
L293-295, How to prove the re-isolated fungi was the same as the inoculated one? Please describe.
Response: Thank you for the observation. We have revised the section and added information on the identification of the re-isolated fungi. The causative agent in the diseased leaf parts was re-isolated on Potato Dextrose Agar (PDA). The morphological characteristics and ITS1/4 sequences of the re-isolates were compared with that of the original cultures.
L312-313, The growth rate of different strains significant varied, please explain the possible reasons and how it will affects pathogenicity test?
Response: Thank you for the comment. We have provided the information under the discussion section.
L348, The NCBI accession number ON764782 is invalid, please update.
Response: Thank you for the observation. We have updated the NCBI accession number and should be ON764382 (W008-L).
- the initial pathogenicity test were use wound/drop inoculation method, but in the greenhouse screening trials were use spray method. The author should discuss more about the difference and possible impact on the outcome between wound and spray inoculation in the discussion section.
Response: We have provided information on the differences of the two method and the possible impacts of each under the discussion section.
Minor comments:
L136 – no humidity shown
Response: Thank you for the observation. We have revised the section and included the value for humidity.
L139 – spore suspension concentration should be use conidia/mL.
Response: Thank you for the observation. We have revised and used conidia/mL.
L140 – The authors should explain why the control is using PDA but not sterile distilled water.
Response: Thank you for noting this. This was a mistake; the negative control we used during pathogenicity assays was sterile distilled water, same as for the greenhouse trials.
L145, L165 - the degrees Celsius should after plus-minus sign.
Response: Thank you for noting this. We have revised the section accordingly.
Round 2
Reviewer 1 Report
1. In Lines 235-236, ‘1 - 3, 3 - 6, 6 - 9 and 9 - 12 weeks’ should be changed into days.
2. The equation of AUDPC was presented two times, please remove the second one.
3. In the title of Table 6, ‘Colletotrichumsublineolum’?
4. In Lines 500-508, ‘Table 7’ is right? Not Table 8? This point should be revised.
5. In Lines 557-558, ‘this method involves o in vivo inoculation and incubation under controlled laboratory conditions, ’? is it right?
Author Response
The authors appreciate the time and effort by the reviewer for reviewing this manuscript and for providing insightful comments and suggestions to improve its quality. We have revised the manuscript as per the reviewers’ suggestions and comments. We have incorporated the following reviewer’s specific comments in the revised version of manuscript.
- In Lines 235-236, ‘1 - 3, 3 - 6, 6 - 9 and 9 - 12 weeks’ should be changed into days.
Response: Thank you for the feedback. We have changed the weeks to days post-inoculation.
- The equation of AUDPC was presented two times, please remove the second one.
Response: Thank you for the observation. We have deleted the AUDPC in section 2.6.2 and in the section referred to the equation described in section 2.6.1.
- In the title of Table 6, ‘Colletotrichumsublineolum’?
Response: Thank you for the observation. We have revised to “Colletotrichum sublineola”
- In Lines 500-508, ‘Table 7’ is right? Not Table 8? This point should be revised.
Response: Thanks for the observation. It was supposed to be Table 8 and not table 7. We have revised the section accordingly.
- In Lines 557-558, ‘this method involves o in vivo inoculation and incubation under controlled laboratory conditions, ’? is it right?
Response: Thank you for the feedback. We have revised the section; it is supposed to be in vitro and not in vivo.
Reviewer 2 Report
Only minor errors listed below:
1. Please double check the misplaced term "concatenated" in L26 (Abstract), L197 and L404 (Fig. 2).
2. The Figure in L361 should be Figure "1".
Author Response
The authors appreciate the time and effort by the reviewer for reviewing this manuscript and for providing insightful comments and suggestions to improve its quality. We have revised the manuscript as per the reviewers’ suggestions and comments. We have incorporated the following reviewer’s specific comments in the revised version of manuscript.
- Please double check the misplaced term "concatenated" in L26 (Abstract), L197 and L404 (Fig. 2).
Response: Thank you for the observation. We have deleted the misplaced term "concatenated" in lines 26, 197 and 404.
- The Figure in L361 should be Figure "1".
Response: Thank you for the observation. We have revised too Figure 1.